# Factors Affecting the Success of Endodontic Microsurgery: A Cone-Beam Computed Tomography Study

**DOI:** 10.3390/jcm11143991

**Published:** 2022-07-10

**Authors:** Daniel Bieszczad, Jaroslaw Wichlinski, Tomasz Kaczmarzyk

**Affiliations:** 1NZOZ Centrum Stomatologii s.c. Justyna Wichlinska, Jaroslaw Wichlinski, ul. 3-go Maja 16, 38-300 Gorlice, Poland; danbiesz@poczta.onet.pl (D.B.); wichlinski@gmail.com (J.W.); 2Department of Oral Surgery, Jagiellonian University Medical College, ul. Montelupich 4, 31-155 Krakow, Poland

**Keywords:** periapical lesion, prognostic factors, endodontic microsurgery, cone-beam computed tomography

## Abstract

The purpose of this retrospective study was to verify preoperative local parameters of periapical lesions evaluated on cone-beam computed tomography (CBCT) scans as a potential prognostic factor in endodontic microsurgery (EMS). Among 89 cases, local factors (dimensions of lesion, bone destruction pattern, presence/absence of cortical bone destruction, height of buccal bone plate, apical extend of root canal filling, presence/absence of communication with anatomical cavities, type of lesion restriction) were measured on preoperative CBCT images before EMS. At least one year after surgery, the outcome of EMS was classified as a success or a failure. Ten cases (11.24%) were classified as a failure and 79 as a success (88.76%). Symptomatic lesions (OR = 0.088 (95% CI 0.011–0.731); *p* = 0.024), apicomarginal lesions (OR = 0.092 (0.021–0.402); *p* = 0.001) and an association with molar teeth (OR = 0.153 (0.032–0.732); *p* = 0.019) were found as negative predictive factors in the univariate analysis, whereas large apicocoronal dimension (OR = 0.664 (0.477–0.926); *p* = 0.016), apicomarginal lesions (OR = 0.058 (0.006–0.55); *p* = 0.013), and an association with molar teeth (OR = 0.047 (0.003–0.869); *p* = 0.04) were identified as negative predictive factors in the multivariate analysis model. Symptomatic lesions, apicomarginal lesions, lesions associated with molar teeth and large apicocoronal dimensions are significantly associated with the failure of EMS.

## 1. Introduction

A few epidemiological studies report that 30–60% of endodontically treated teeth still show apical periodontitis [1,2]. It may be related to primary or secondary infection of the root canal, extraradicular infections, cementum tears or microcracks/fractures of the root. Endodontic microsurgery (EMS) is a surgical procedure which aims to remove persistent periapical lesions of endodontically treated teeth and also to regenerate periapical tissues. Currently, with the development of treatment techniques, EMS may be performed with the use of magnification, high power illumination, ultrasonic root-end preparation, and advanced types of biocompatible materials. The success rate of EMS has been reported to be approximately 80–90% [2,3,4,5]. Even microsurgical re-treatment shows a high success rate [6]. Relatively recently introduced cone-beam computed tomography (CBCT) has had a major impact on increasing the success rate of EMS. It is used for the presurgical assessment of the extent of the inflammatory lesion, ascertaining the relationship between the lesion and adjacent anatomical structures, determining the cause of the disease itself and also identify local parameters that can serve as potential prognostic factors in EMS.

Regardless of the development of materials, equipment and procedure techniques, there is still deficiency in evidence-based data regarding local factors affecting the success of EMS [7]. Determining prognostic factors may have a decisive influence on the type of treatment [8] or the procedure technique [9]. They are divided into three categories: patient-related, tooth-related, and treatment-related factors [7,10]. Most of the extensive research in which patient-related factors were studied has focused on the relationship between age/gender and the outcome of apical surgery [10,11,12,13,14,15]. In turn, various studies on tooth-related factors have considered mainly the type of tooth, demonstrating significantly higher success rate for EMS regarding the anterior teeth [10,11,16,17,18,19], as well as the size of preoperative periapical lesion [20,21]. Most authors have assessed its dimensions during the surgical procedure [20,22]; however, due to the fact that they were taken after root-end resection and enucleation of pathologic tissues, the dimensions were larger relative to its actual size [23]. By contrast, CBCT enables very accurate calculations, yet investigations employing CBCT for measurement of periapical lesions are scarce [23,24]. Furthermore, the results of the studies evaluating the association of the dimension of a lesion with EMS outcome are ambiguous and difficult to compare, as some researchers employ guided tissue regeneration [10], while others do not [23]. In addition, there is no consensus on whether the height of buccal bone plate [20,21,23], the type of outline of lesion [15,21] and the destruction of buccal or palatal cortical bone [21,23,24] have any effect on the outcome of EMS. These factors, however, should be thoroughly investigated as they can be accurately measured with the use of CBCT and a detailed understanding of their impact on the outcome of EMS can increase the success rate, by excluding the cases with poor prognosis.

Among the treatment-related factors, one of the most well-studied is a root-end filling material. In this respect, the best outcomes were obtained for advanced biocompatible materials, such as MTA+ (Cerkamed) or Biodentine^®^ (Septodont) [10,11,25]. In addition, some studies showed the advantage of primary EMS over secondary procedure [10,12], and better results with the use of ultrasonic micro-tips over burs [11,19,26]. The effect of GTR on the outcome of EMS is uncertain as the results of some studies indicate its beneficial influence in “apicomarginal” and “through and through” lesions [24,27,28], whereas other report the absence of any effect, especially in lesions with bone defects confined to the periapical region [29,30]. Similarly, regarding EMS of “large lesions > 10 mm”, the results of some reports linked better outcomes with EMS involving GTR [31]; however, studies with opposite conclusions are also available [32].

Thus, the aim of this retrospective study was to estimate local parameters of preoperative periapical lesions evaluated on CBCT scans as a potential prognostic factor in EMS.

## 2. Materials and Methods

With the approval no 1072.6120.192.2021 by the Bioethical Committee of the Jagiellonian University, Krakow, Poland, clinical cases for the present analysis were retrospectively collected from a database of patients who received EMS at a single dental center (NZOZ Centrum Stomatologii s.c. Justyna Wichlinska, Jaroslaw Wichlinski, Gorlice, Poland) between March 2015 and June 2020. Within this period, 89 teeth in 80 patients were subjected to EMS and were thus included in this analysis.

Inclusion criteria were teeth with persistent periapical lesions after endodontic treatment, preoperative CBCT, and follow-up of at least one year after surgery. Teeth with fractures or cracks and the teeth of patients under the age of 18 were excluded from this analysis. Before surgery, 76 teeth underwent endodontic treatment in our office, out of which 37 underwent primary endodontic treatment (non-vital teeth involved with a lesion), 39 underwent retreatment (subsequent to unsuccessful initial root canal therapy), and 13 were not treated endodontically in our clinic. Teeth were not subjected to the secondary endodontic treatment in case of resorption of the apex, large post with thin root canal walls, perforations or incorrect position of post (with high-risk of perforation), or in case of patient’s disagreement to the removal of prosthetic reconstruction. Follow-up visits after endodontic treatment (both primary and secondary) took place after 6–12 months. In the event of a lack of radiological signs of healing, the lesion was classified as persistent and submitted to EMS. Only in cases where the lesion involved two or more teeth and acute signs of inflammation (e.g., recurrent abscesses) did patients qualify for immediate surgery. All surgeries were performed by one dentist (D.B.), using a standardized microsurgical technique utilizing an operating microscope (Leica M320, Leica Microsystems, Heerbrugg, Switzerland) and Piezosurgery^®^ (Mectron, Carasco, Italy). Biocompatible materials such as MTA+ (Cerkamed, Stalowa Wola, Poland) or Biodentine^®^ (Septodont, Saint-Maur-des-Fossés, France) were used as a root-end canal filling with the use of MAP system^®^ (Produits Dentaires SA, Vevey, Switzerland) [Figure 1]. The operating microscope was used to inspect the surface of the root apex after resection to find lateral and additional canals or isthmuses and for retrograde filling. Piezosurgery^®^ was used to enucleate the pathological tissue, with the use of ultrasonic micro-tips for preparing root-end cavity. In some apicomarginal, “through and through” and > 10 mm lesions GTR were used with no protocol (owing to the financial limitations of some patients). The following factors were calculated on presurgical CBCT scans:Apicocoronal maximum diameter (measured in the sagittal section) (Figure 2);Buccolingual maximum diameter (measured in the sagittal section) (Figure 3);Mesiodistal maximum diameter (measured in horizontal section) (Figure 4);Bone destruction pattern (rated in all sections);Presence/absence of cortical bone destruction;Height of buccal bone plate (measured in sagittal section);Apical extent of root canal filling;Presence/absence of communication with anatomical cavities;Type of lesion restriction (rated as diffused or demarcated).

The volume of lesions was calculated using ITK-SNAP (free software under the GNU General Public License) [33,34]. Details on the calculation of the volume of the lesions are given in Figure 5, Figure 6 and Figure 7.

Additionally, patient’s age and gender, tooth position, activity of the lesion (symptomatic/asymptomatic) and time elapsed from surgery to follow-up were collected. All factors were measured independently by two examiners (D.B and J.W).

The follow-up visit took place at least one year after EMS. To evaluate the outcome of EMS, clinical and radiographic records were analyzed. Clinical evaluation included the assessment of any of the following signs and symptoms: tenderness on palpation or percussion, loss of function, tooth mobility, periodontal pocket, and sinus tract formation. Radiographic healing was evaluated on CBCT images (based on modified PENN 3D Criteria) [33] or on periapical radiographs (insofar as it was taken along with CBCT before surgery) according to the criteria by Rud [35] and Molven [36]. Following the discussion of the criteria for calibration of assessment, radiographic healing was evaluated independently by two examiners (D.B. and J.W.) and classified as: complete, incomplete, uncertain or unsatisfactory healing [33,35,36]. In turn, the outcome was classified as a success or failure. The success was ascertained when radiographic healing was classified as “complete healing” or “incomplete healing” and when during the follow-up period (at least 12 months) no clinical signs or symptoms (tenderness on palpation or percussion, loss of function, tooth mobility, periodontal pocket, sinus tract formation) were recorded. Failure was ascertained when radiographic healing was graded as “uncertain healing” or “unsatisfactory healing” and any of the clinical signs or symptoms were confirmed during the follow-up period. In cases where the GTR was used (apicomarginal, “through and through” and > 10 mm lesions), its effect on the treatment outcome was also analyzed.

All CBCT images were taken using the same unit CS 8100 3D (Carestream Dental^®^) at a resolution of 150 microns. The unit generated DICOM files (Digital Imaging and Communications in Medicine), which were imported into the Carestream Dental Imaging Software and evaluated on a 25-inch medical monitor independently by two calibrated examiners (D.B. and J.W).

All statistical analyses were performed using the R Project for Statistical Computing [version 4.1.0] [37]. The inter-examiner agreement of quantitative variables in the preoperative CBCT was assessed by calculating the interclass correlation coefficient (ICC2) for absolute agreement with a 2-way random effects [38], and—in the case of qualitative variables with the Cohen kappa statistic [39]. The inter-examiner agreement of postoperative outcomes was also assessed with the Cohen kappa coefficient. The interpretation of agreements for ICC2 was made in accordance with Fleiss and Shrout [40] and, for the kappa value, in accordance with McHugh [41].

Uni- and multivariate analyses describing the influence of variables on dichotomous outcome (success/failure) were performed utilizing the logistic regression model. Results were presented as OR (Odds Ratio) with 95% confidence interval (CI). The level of significance was set at 0.05

## 3. Results

Complete data sets of 89 roots were analyzed statistically. Table 1 presents the distribution of cases according to the preoperative parameters and follow-up period. The overall success rate was 88.76% (79/89).

The mean ICC2 for quantitative variables was 0.99 (95% CI 0.994–0.999), and for qualitative variables the mean kappa value was 0.947 (95% CI 0.702–1), which shows excellent agreement between the two examiners (D.B. and J.W.). Table 2 shows the distribution of cases according to the tooth-related parameters.

In univariate logistic regressions, molar teeth, preoperative activity of a lesion and apicomarginal lesions were the only statistically significant predictors of the treatment failure (Table 3).

By employing the multivariate logistic regression model, we found that molars, apicomarginal lesions and apicocoronal maximum diameter were significantly associated with a negative outcome (Table 4).

The results of the univariate logistic regression model regarding treatment of apicomarginal, through and through, and lesions of volume >475.5 mm^3^ with the use of GTR are presented in Table 5. None of the subgroups reached the level of statistical significance.

## 4. Discussion

The present retrospective study evaluated preoperative variables as possible predictors of the healing outcome of EMS. These are decisive factors as the necessity of EMS can be weighed against alternative methods of treatment (e.g., hemisection, tooth extraction). We utilized CBCT scans for the measurement of tooth-related factors as they provide a more detailed image, since the results of several studies have demonstrated high conformity between CBCT measurements and actual intraoperative measurements in contrast to periapical radiographs [34,42,43]. The excellent correlation between examiners in the evaluation of linear measurements in our study shows the ease of interpretation and repeatability of the assessment of periapical lesions on CBCT scans.

The results of the current series (based on univariate logistic regression) indicate that tooth position, preoperative activity of a lesion, and apicomarginal location of lesions had a significant effect on the healing outcome. Other factors did not influence the outcome of EMS; however, the maximum apicocoronal dimension was shown to be a significant factor in the multivariate analysis model.

Among all analyzed factors, apicomarginal location of a lesion had the most significant impact on the healing outcome. In some of the previous reports, the authors excluded such lesions due to the expected adverse effect on the outcome [15], and also due to difficulty of deciding the appropriate boundary of the periapical lesions faced by the examiners [23]. Our linear inter-examiner reliability of factor measurements showed very high agreement, therefore we decided to include apicomarginal lesions in the current analysis.

In the current series, there were significant differences in the treatment outcome between apical and apicomarginal lesions. These observations are consistent with the results of Jean Nee Lui et al. [9], Song et al. [21,44] and Kim et al. [45], who also found a better outcome for the apical lesions. The loss of alveolar bone induces apical migration of gingival epithelial cells, forming a long junctional epithelium, which makes periodontal reattachment difficult [9]. One may speculate that microcracks, sometimes undetectable even on CBCT scans or with the use of microscope, might also account for worse results of healing for apicomarginal lesions. Even so, we believe that it is worth trying to treat such lesions, especially with the use of GTR, and although we failed to demonstrate its significant impact on the outcome, the current evidence suggests that GTR substantially improves the results of EMS in the case of apicomarginal lesions [27,44] as it boosts periapical and periodontal healing, increases both the amount of bone and cementum formation, and prevents apical migration of the junctional epithelium into the periodontal pocket by acting as a barrier for unwanted cells such as epithelial cells [32].

Another factor that we found to have a significant impact (both in univariate and multivariate logistic regression) on the outcome of EMS was the molar group of teeth. Anterior teeth and premolars had a success rate of over 90%, in contrast to molars with a mere 60% successful outcome. Similar results were demonstrated in most of the published data [7,15,44]. This is most likely due to the complexity of canal anatomy (isthmuses, additional canals) and the difficulty in accessing molar root apices; it has been shown that even a surgical operating microscope does not enhance accessibility [46]. Some studies, however, reported no effect of type of teeth on the surgical treatment outcome, but the authors indicated that this might have stemmed from the unequal distribution of cases within their groups [9,14]. Therefore, the clinician should pay special attention when subjecting molars to apical microsurgery (mouth opening width, proximity to the roots of adjacent teeth, anatomical bone prominences) and also consider some other treatment options (retreatment or tooth extraction).

In the current study, EMS of teeth with preoperative pain, sinus tract, and tenderness on palpation had significantly worse results in comparison with asymptomatic teeth. Other authors also demonstrated poorer results in the treatment of preoperatively symptomatic lesions [7,10,11,47]. Acute stage of infection may compromise the healing potential of the surgical wound [7]. Moreover, clinical signs of exacerbation can have a deleterious effect on the regeneration process after surgery, as they have been associated with extra-radicular infections, in particular with actinomycosis [47,48]. EMS might not always completely eradicate these bacteria and, hence, the risk of subsequent extra-radicular reinfection [7]. Interestingly, Jeen-Nee Lui et al. [9] and Song et al. [15] found no significant differences in terms of the outcome between presurgically symptomatic and asymptomatic lesions. Song et al. [15] suggested that the use of transillumination and magnification in endodontic microsurgery might allow for complete eradication of the source of reinfection regardless of tooth status. In our opinion it is indeed possible with regard to small, well-demarcated lesions, but rather hard to achieve in larger lesions with destroyed bone plates.

The dimensions of a lesion is another factor that has been considered by most authors. Our results suggest that none of the dimensions (apicocoronal, buccolingual, mesiodistal) had any significant influence on the outcome of EMS in univariate logistic regression (though apicocoronal size almost reached the level of significance); however, the maximum apicocoronal dimension proved to be a significant factor in the multivariate analysis. Presumably, in lesions with high apicocoronal diameter, a significant part of pathological tissues is located behind the roots, and this area is difficult to access for complete curettage. Other studies that addressed the issue of a lesion’s dimensions produced divergent results. Von Arx et al. [10] demonstrated better prognosis for lesions with a apicocoronal size below 5 mm, whereas Kim et al. [23] found no significant differences in the outcome between groups of over or below 6 mm. With reference to the buccolingual size, we did not find a significant association with the outcome of EMS, which is in line with the results of Kim et al. [23]. Interestingly, von Arx et al. [20] showed better healing in the case of lesions with buccolingual maximum dimensions below 7.15 mm. However, their measurements were taken intraoperatively after bone preparation, thus it is difficult to compare them directly with our findings. In turn, the results of our study confirm the conclusions of Kim et al. [23], who demonstrated that mesiodistal size did not influence the outcome of EMS; however, a study by von Arx et al. [20] suggested better healing in the case of lesions where the length of access window to a bony crypt (measured intraoperatively) was below 7.04 mm. Given that healing occurs within the postoperative bone defect, the postoperative dimension of the lesion could have a larger impact on the prognosis than the preoperative one [23]. Thus, further research into the correlation of the preoperative radiographic size of lesions with the dimensions of the intraoperative bony crypt after bone removal, curettage, and root-end resection is necessary.

Although the mean volume of lesions in the current study was very high (475 mm^3^), we failed to demonstrate its impact on the treatment outcome. While some authors did come to similar conclusions [49,50], Von Arx et al. [20] demonstrated better results in the case of lesions with a volume <395 mm^3^. In turn, Kreisler et al. [22] and Kim et al. [23] showed better healing with lesions in which the volume was below 60 mm^3^ and 50 mm^3^, respectively. Apart from the current study, only Kim et al. [23] assessed the preoperative volume of lesions using CBCT scans, but they employed a different method to calculate the lesion’s volume. They determined it by summing the volumes of all slices, which were automatically calculated after manual outlining with the use of the sculpting tool in their software. On the other hand, von Arx et al. [20] measured the volume of a lesion intraoperatively after bone preparation by multiplying the length, the depth and the height of a bony crypt. Worse prognosis for larger lesions in some studies might be related with the fact that smaller apical lesions require more extensive osteotomy to gain access to a lesion, resulting in complete eradication of the pathologic tissue and microorganisms. Additionally, a “fresh” osseous wound created by surgical enlargement of small lesions may facilitate bone formation [7]. Caliscan et al. [51] suggested that the curettage of large lesions may be incomplete and residual tissue may act as a source of persistent inflammation. We did not, however, find that relationship, which may be related to the use of piezosurgery, which uses piezoelectric ultrasonic vibrations to perform precise osteotomies, and more effective debridement. This may also account for the lack of statistical differences in healing outcome between larger and smaller lesions in the current series. To the best of our knowledge, no previous study tested the effect of piezosurgery on the outcome of EMS.

It is no surprise that the destruction of both cortical plates (“through and through” lesions) had poorer outcomes (75%) than the lesions with no destruction of any plate (87%), or with the destruction of only one plate (91.84%). However, the differences were not significant. The periosteum is believed to play a very important role as a source of osteocomponent cells and serves as a barrier against epithelial cell migration into the healing sites [31]. Large bony defects and “through and through” lesions in which both cortical bone plates are lost result in likely periosteum damage by the inflammatory process. Consequently, extensive periapical bony destruction tends to be healed with fibrous connective tissue [28], and healing appears to require more time. In our practice, we try not to convert lesions with the destruction of one plate into “through and through” lesions by changing the typical surgical access. This mainly applies to the lateral incisors with the destruction of the palatal plate. In such instances, we try to use the palatal access instead of the buccal; however, it is a difficult procedure, and the adequate local conditions must be met (small distance between root apex and palatal plate, high roof of a palate). The results of our study suggest that the use of GTR in “through and through” lesions might be of some benefit in comparison with no bone regeneration; however, we failed to demonstrate any significant impact of its use. Nevertheless, previous studies did demonstrate the relevance of the application of bone grafts with collagen membranes in treatment of “through and through” lesions [21,24], and showed that in cases without GTR, postoperative CBCT scans revealed scar tissue formation as a “bone tunnel”, in comparison with images of the full bone regeneration following the use of GTR [21,52].

The height of the buccal bone plate may be potentially considered as another factor impacting the treatment outcome. Most studies [9,20,23,49] did not, however, demonstrate its significance, which is in line with the current results. By contrast, Song et al. [21] showed a better outcome (94.3% vs. 68.8%) for lesions with the height of buccal plate >3 mm. Likewise, Wesson and Gale [53] found better healing of lesions with bone cuff of 3 mm (76%) compared to lesions with bone cuff of 1 mm (47%). The differences in outcomes reported in studies might be explained by various surgical techniques and the use of various healing criteria [20]. Accordingly, during endodontic microsurgical procedures, buccal bone plate height of at least 3 mm or more should be secured [21].

In our study, both types of lesions (isolated and with communication with anatomical cavities) had comparable results, with no statistical significance, just like the group with demarcated and diffused lesions. No type of root canal filling reached the level of statistical significance. Additionally, the gender and age differences in terms of healing after apicoectomy were not statistically significant in our study.

The minimum period of time between surgery and follow-up visit was established as 12 months, which is commonly adopted in most studies [9,10,14]. The mean follow-up was 2.75 ± 1.33 years. A long-term study by Jesslen et al. [54] with a 5-year follow-up demonstrated that assessing the outcome after 1 year was valid in more than 95% of the cases. On the other hand, Grung et al. [49] in long term follow-up showed an overall success rate of 87.2%, which was higher than 80.9% at the follow-up appointment after one year. In previous studies, there appears to have been a tendency for cases to transition from incomplete to complete healing, or uncertain to incomplete or unsatisfactory healing over time. In particular, larger lesions needed more time to heal, but after one year, an examiner can qualify a case into success or failure.

In the current series, all surgical procedures were carried out by one operator. This can be considered either as a strength or a weakness of the study. The strengths of this study, due to the minimalization of intraoperative variations, relate to the surgical procedure [23]. The weaknesses are due to the small study population, which cannot represent the generalized outcome of EMS [23]. Additionally, in our opinion, technical habits and unintentional mistakes related to having only a single operator might be repeated, and this can have an impact on the outcome. Thus, understanding the influence of individual factors on the healing process is important. The success rate of all surgeries in this series was almost 89%, which is comparable with other studies.

Another limitation of this study is the small study sample and the lack of protocol when using GTR. Thus, current results require confirmation in larger-scale studies with a straightforward GTR protocol.

## 5. Conclusions

To sum up, EMS is a very effective procedure for the treatment of periapical lesions. Worse outcomes may be associated with molar teeth, apicomarginal lesions, preoperatively symptomatic lesions and lesions of large apicocoronal dimension. Cases with those characteristics should be treated with great caution, and the detailed planning of the surgical procedure should be based on additional data from CBCT images (e.g., perforations, lateral canals, root cracks or fractures). Furthermore, a patient should be informed in advance about a worse prognosis. Cone-beam computed tomography should be a mandatory diagnostic step before EMS.

## Figures and Tables

**Figure 1 jcm-11-03991-f001:**
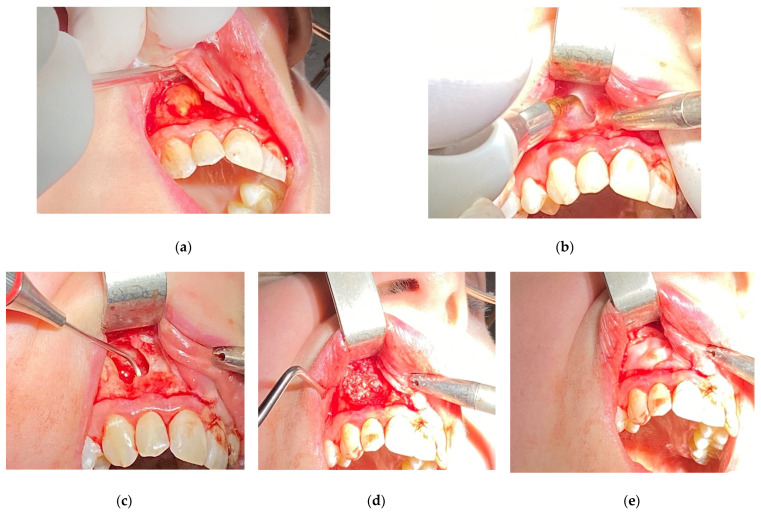
EMS case presentation (**a**–**e**): (**a**) exposure of the periapical lesion following elevation of the mucoperiosteal flap; (**b**) effective debridement of the bone defect with the use of Piezosurgery^®^ (Mectron); (**c**) filling the root-end cavity with the use of MAP system^®^ (Produits Dentaires SA); (**d**) filling of the defect with bone substitute graft; (**e**) resorbable membrane covering the graft.

**Figure 2 jcm-11-03991-f002:**
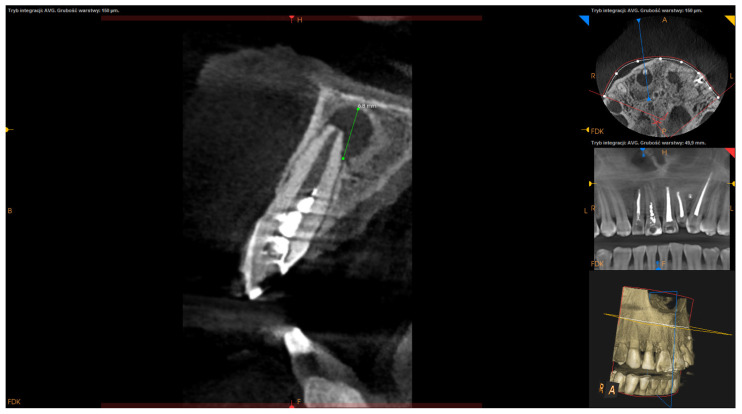
CBCT image (CS 8100 3D Carestream Dental^®^) measurement of apicocoronal maximum diameter in sagittal section. Non-English annotations present software technicalities (AVG integration mode and the layer thickness).

**Figure 3 jcm-11-03991-f003:**
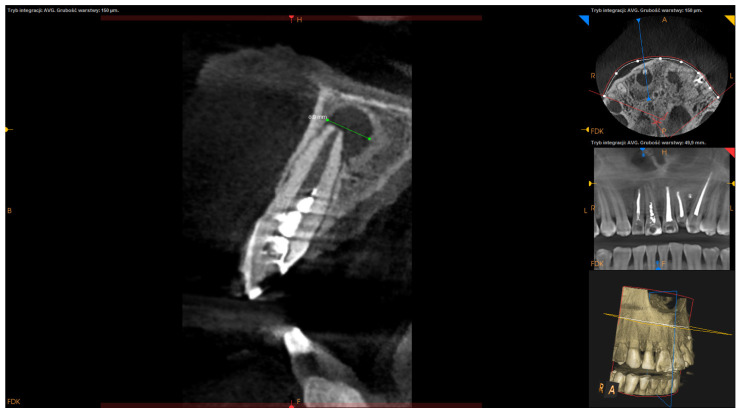
CBCT image (CS 8100 3D Carestream Dental^®^) measurement of buccolingual maximum diameter in sagittal section. Non-English annotations present software technicalities (AVG integration mode and the layer thickness).

**Figure 4 jcm-11-03991-f004:**
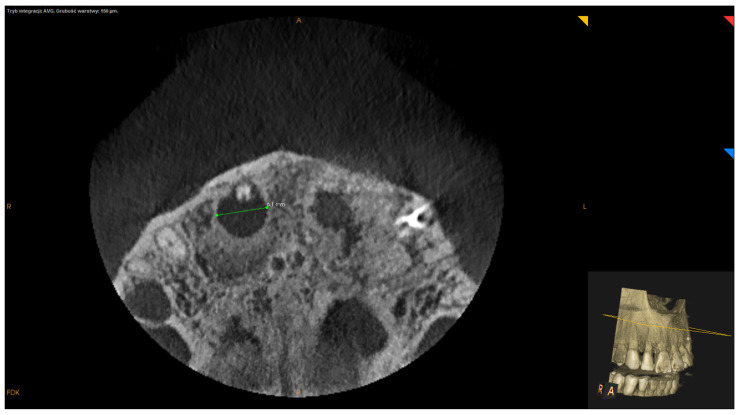
CBCT image (CS 8100 3D Carestream Dental^®^) measurement of mesiodistal maximum diameter in horizontal section. Non-English annotations present software technicalities (AVG integration mode and the layer thickness).

**Figure 5 jcm-11-03991-f005:**
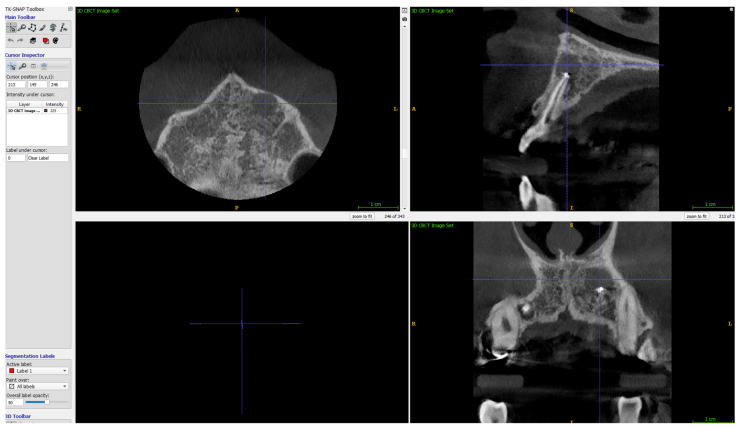
ITK-SNAP program surface for segmentation of 3D lesions, in saggital, axial, and coronal views. The defect is marked with the cursor, before manual limiting the field of interest.

**Figure 6 jcm-11-03991-f006:**
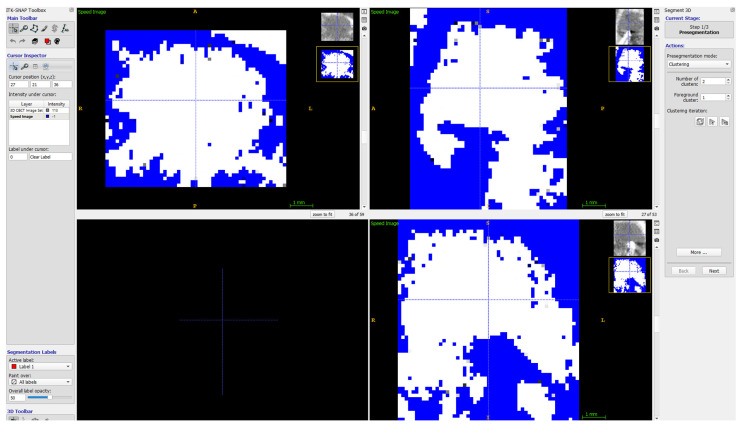
ITK-SNAP the view after semiautomatic defect recognition, before initiation of the semiautomatic algorithm using spherical fillers “bubbles”, filling the “white” area.

**Figure 7 jcm-11-03991-f007:**
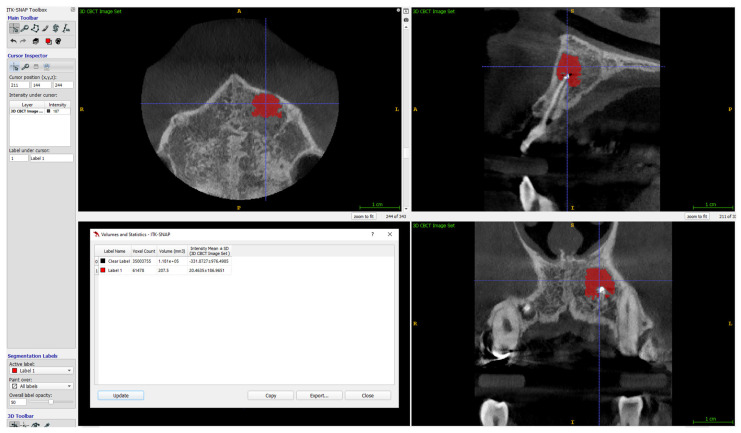
ITK-SNAP completed segmentation and visualisation of a lesion, with the volume result (after direct expression of the number of voxels and conversion into mm^3^).

**Table 1 jcm-11-03991-t001:** The descriptive statistics of the study sample.

Parameter		*N* = 89
Age (years)	mean ± SD	44.25 ± 10.71
	median (range)	45 (37–52)
Gender	F	45 (50.56%)
	M	44 (49.44%)
Follow-up period [years]	mean ± SD	2.75 ± 1.33
	median (range)	3 (2–4)
Preoperative activity of the lesion	symptomatic	44 (49.44%)
	asymptomatic	45 (50.56%)
Teeth group	anterior	54 (60.67%)
	premolars	25 (28.09%)
	molars	10 (11.24%)

**Table 2 jcm-11-03991-t002:** Distribution of cases according to the tooth-related parameters (continuous variables are split at the mean values).

Parameter		*N* = 89
volume of the lesion M ± SD: 475.51 ± 680.33	<475.5 mm^3^	66 (74.16%)
	>475.5 mm^3^	23 (25.84%)
apicocoronal maximum diameter M ± SD: 8.93 ± 3.71	<8.93 mm	48 (53.93%)
	>8.93 mm	41 (46.07%)
mesiodistal maximum diameter M ± SD: 8.86 ± 4.65	<8.86 mm	54 (60.67%)
	>8.86 mm	35 (39.33%)
buccolingual maximum diameter M ± SD: 6.91 ± 2.43	<6.91 mm	48 (53.93%)
	>6.91 mm	41 (46.07%)
bone destruction pattern	apical	68 (76.40%)
	apicomarginal	21 (23.60%)
cortical bone destruction	none	32 (35.96%)
	destruction of one of the cortical plates	49 (55.06%)
	destruction of both cortical plates (through and through lesions)	8 (8.99%)
height of buccal bone plate	<3 mm	10 (11.24%)
	>3 mm	79 (88.76%)
apical extend of root canal filling	0–2 mm short of apex	43 (48.31%)
	>2 mm short of apex	16 (17.98%)
	beyond apex	30 (33.71%)
communication with anatomical cavities	present	13 (14.61%)
	absent (isolated lesions)	76 (85.39%)
type of lesion restriction	diffused	44 (49.44%)
	demarcated	45 (50.56%)

M, mean, SD, standard deviation.

**Table 3 jcm-11-03991-t003:** Distribution of cases per preoperative demographic and clinical data as well as radiographic factors with the success rate; univariate logistic regression model.

Parameter		Success	OR (95% CI)	*p*-Value Univariate Logistic Regression
gender	female (*N* = 45)	41 (91.11%)	1.00	
	male (*N* = 44)	38 (86.36%)	0.618 (0.162–2.36)	0.481
age [years]			0.939 (0.877–1.006)	0.072
preoperative activity of a lesion	non-active (*N* = 45)	44 (97.78%)	1.00	
	active (*N* = 44)	35 (79.55%)	0.088 (0.011–0.731)	0.024 *
teeth group	anterior (*N* = 54)	49 (90.74%)	1.00	
	premolars (*N* = 25)	24 (96.00%)	2.449 (0.271–22.133)	0.425
	molars (*N* = 10)	6 (60.00%)	0.153 (0.032–0.732)	0.019 *
follow-up period	years		0.911 (0.556–1.492)	0.71
apicocoronal maximum diameter [mm]			0.85 (0.721–1.002)	0.053
	<8.93 (*N* = 48)	45 (93.75%)	1.00	
	>8.93 (*N* = 41)	34 (82.93%)	0.324 (0.078–1.345)	0.121
mesiodistal maximum diameter	mm		0.953 (0.835–1.088)	0.475
	<8.86 (*N* = 54)	49 (90.74%)	1.00	
	>8.86 (*N* = 35)	30 (85.71%)	0.612 (0.164–2.292)	0.466
buccolingual maximum diameter	mm		0.824 (0.642–1.057)	0.128
	<6.91 (*N* = 48)	44 (91.67%)	1.00	
	>6.91 (*N* = 41)	35 (85.37%)	0.53 (0.139–2.027)	0.354
volume of a lesion	mm		1 (0.999–1)	0.435
	<475.5 (*N* = 66)	59 (89.39%)	1.00	
	>475.5 (*N* = 23)	20 (86.96%)	0.791 (0.187–3.353)	0.75
bone destruction pattern	apical (*N* = 68)	65 (95.59%)	1.00	
	apicomarginal (*N* = 21)	14 (66.67%)	0.092 (0.021–0.402)	0.001 *
cortical bone destruction	none (*N* = 32)	28 (87.50%)	1.00	
	destruction of one of the cortical plates (*N* = 49)	45 (91.84%)	1.607 (0.372–6.948)	0.525
	destruction of two plates (*N* = 8)	6 (75.00%)	0.429 (0.063–2.902)	0.385
height of buccal bone plate	<3 mm (*N* = 10)	8 (80.00%)	1.00	
	>3 mm (*N* = 79)	71 (89.87%)	2.219 (0.4–12.307)	0.362
type of lesion restriction	diffused (*N* = 44)	37 (84.09%)	1.00	
	demarcated (*N* = 45)	42 (93.33%)	2.649 (0.638–10.989)	0.18
communication with anatomical cavities	absent (isolated lesions) (*N* = 76)	68 (89.47%)	1.00	
	present (*N* = 13)	11 (84.62%)	0.647 (0.121–3.455)	0.611
apical extend of root canal filling	0–2 mm short of apex (*N* = 43)	39 (90.70%)	1.00	
	>2 mm short of apex (*N* = 16)	12 (75.00%)	0.308 (0.067–1.42)	0.131
	beyond apex (*N* = 30)	28 (93.33%)	1.436 (0.246–8.392)	0.688

Asterisk denotes significance; OR, Odds Ratio, CI, confidence interval.

**Table 4 jcm-11-03991-t004:** Multivariate logistic regression model.

Parameter		OR (95% CI)	*p*-Value Multivariate Logistic Regression
preoperative activity of a lesion	non-active	1.00	
	active	0.139 (0.011–1.721)	0.124
tooth group	anterior	1.00	
	premolars	0.924 (0.064–13.314)	0.954
	molars	0.047 (0.003–0.869)	0.04 *
bony destruction pattern	apical	1.00	
	apicomarginal	0.058 (0.006–0.55)	0.013 *
age	years	0.919 (0.821–1.029)	0.143
apicocoronal maximum diameter	mm	0.664 (0.477–0.926)	0.016 *

Asterisk denotes significance; OR, Odds Ratio, CI, confidence interval.

**Table 5 jcm-11-03991-t005:** Distribution of cases with indications for GTR per outcome; univariate logistic regression model.

Parameter		Success Rate	OR (95% Cl)	*p*-Value Univariate Logistic Regression
apicomarginal lesions	without GTR (*N* = 14)	7 (50.00%)	1.00	
	with GTR (*N* = 7)	6 (85.71%)	6 (0.565–63.676)	0.137
lesions of volume >475.5 mm^3^	without GTR (*N* = 10)	9 (90.00%)	1.00	
	with GTR (*N* = 13)	11 (84.62%)	0.611 (0.047–7.88)	0.706
“through and through” lesions	without GTR (*N* = 3)	2 (66.67%)	1.00	
	with GTR (*N* = 5)	4 (80.00%)	2 (0.078–51.593)	0.676

OR, Odds Ratio, CI, confidence interval.

## Data Availability

The data that support the findings are available on request from the first author, D.B.

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
