# Peer review of "Factors Affecting the Success of Endodontic Microsurgery: A Cone-Beam Computed Tomography Study"

_jcm, 2022, doi:10.3390/jcm11143991_

Round 1
Reviewer 1 Report
This manuscript retrospectively evaluated the factors affecting the success of endodontic microsurgery using the cone-beam computed tomography analysis. The study was well conducted and relevant to the journal. Some of the minor queries have been addressed below.
- Mention how was sample size of 89 was estimated.
- Mention the complete manufacturer’s details of the materials used.
- Instead of the term, “retrograde”, “root end” can be mentioned.
- Whether the parameters involved in root canal treatment like canal enlargement size, irrigants and intracanal medicaments used, type of obturating material and sealer used, and type of obturating technique used were considered in the evaluation?
- The effect of bone graft to fill the bony defect after periapical surgery was considered in this study?
- Mention the statistical analysis used.
***********************************
Author Response
Detailed response to reviewers’ comments
We thank the reviewers for their careful read and thoughtful comments on previous draft. We have carefully taken their comments into consideration in preparing our revision, which has resulted in a paper that is clearer and more compelling.
The following summarizes how we responded to reviewer #1 comments.
Comments of the reviewer #1
- 1. Mention how was sample size of 89 was estimated
Our response:
We now specified that the number of 89 was the total sum of patients subjected to endodontic microsurgery within the period between March 2015 and June 2020 in the dental centre.
Revised text:
By the approval no 1072.6120.192.2021 by the Bioethical Committee of the Jagiellonian University, Krakow, Poland, clinical cases for the present analysis were retrospectively collected from database of patients who received EMS at single dental centre (NZOZ Centrum Stomatologii s.c. Justyna Wichlinska, Jaroslaw Wichlinski, Gorlice, Poland) between March 2015 and June 2020. Within this period, eighty-nine teeth weresubjected to EMS were included into this analysis.
- Mention the complete manufacturer’s details of the materials used.
Our response:
We now specified the complete manufacturer’s details of the material used.
Revised text:
All surgeries were performed by one dentist (D.B.), in standardized microsurgical technique utilizing operating microscope (Leica M320, Leica ) and Piezosurgeryâ“¡ (Mectron). Biocompatible materials such as MTA+ (Cerkamed) or Biodentineâ“¡(Septodont) were used as a root-end canal filling with the use of MAP systemâ“¡ (Produits Dentaires SA) [Fig.1].
- Instead of the term, “retrograde”, “root end” can be mentioned.
Our response:
The term “retrograde” has now been changed to “root end”
Revised text:
Biocompatible materials such as MTA+ (Cerkamed) or Biodentineâ“¡ (Septodont) were used as a root- end canal filling with the use of MAP systemâ“¡ (Produits Dentaires SA) [Fig.1].
[…] Piezosurgeryâ“¡ (Mectron) was used to enucleate the pathological tissue, and with the use of ultrasonic micro-tips for preparing root-end cavity.
- Whether the parameters involved in root canal treatment like canal enlargement size, irrigants and intracanal medicaments used, type of obturating material and sealer used, and type of obturating technique used were considered in the evaluation?
Our response:
These factors were not considered in the current analysis, as we put emphasis on factors that can be measured on preoperative cone-beam computed tomography images. We do agree, however, that they play important role as factors influencing the success of endodontic treatment and we aim to investigate them in the next paper. We have now added required information on irrigants, intercanal medicaments, obturating material and the sealer used in the study.
Revised text:
All endodontic treatments in our clinic were carried out with the use of operating microscope (Leica M320), irrigation with 5,25% NaOCl, rotary instrumentation (ProTaper Next) and vertical condensation of gutta-percha (BeeFill, VDV) with AhPlus (Dentsply Sirona) as a sealant.
- The effect of bone graft to fill the bony defect after periapical surgery was considered in this study?
Our response:
The effect of bone graft to fill the bony defect after periapical surgery was indeed considered in this study. In some apicomarginal, „through and through” and >10mm lesions GTR was used in EMS but with no protocol (owing to financial limitations of some patients). The results (univariate logistic regression) are presented in Table 5. We failed to demonstrate its significant impact on the outcome (although the results were better with the use of GTR).
- Mention the statistical analysis used.
Our response:
All statistical analyses were performed using the R Project for Statistical Computing [version 4.1.0]. The inter-examiner agreement of quantitive variables in the preoperative CBCT was assessed by calculating the interclass correlation coefficient (ICC2) for absolute agreement with a 2-way random effects, and - in the case of qualitative variables - with the Cohen kappa statistic. The inter-examiner agreement of postoperative outcomes was also assessed with the Cohen kappa coefficient. The interpretation of agreements for ICC2 was made in accordance with Fleiss and Shrout and - for the kappa value - in accordance with McHugh. Uni- and multivariate analyses describing the influence of variables on dichotomous outcome (success/failure) were performed utilizing logistic regression model. Results were presented as OR (Odds Ratio) with 95% confidence interval (CI). The level of significance was set at 0.05.
Reviewer 2 Report
Dear Authors
The topic is interesting, however more details should be added along the text. For comments, please see the attached file.

Author Response
Detailed response to reviewers’ comments
We thank the reviewers for their careful read and thoughtful comments on previous draft. We have carefully taken their comments into consideration in preparing our revision, which has resulted in a paper that is clearer and more compelling.
The following summarizes how we responded to reviewer #2 comments.
Comments of the reviewer #2
- Comment no 1
Our response:
“Bony” has now been corrected to “bone” in the text
- Specify how many patients
Our response:
It now has been corrected and specified.
Revised text:
By the approval no 1072.6120.192.2021 by the Bioethical Committee of the Jagiellonian University, Krakow, Poland, clinical cases for the present analysis were retrospectively collected from database of patients who received EMS at single dental centre (NZOZ Centrum Stomatologii s.c. Justyna Wichlinska, Jaroslaw Wichlinski, Gorlice, Poland) between March 2015 and June 2020. Within this period, 89 teeth in 80 patients were subjected to EMS and were thus included into this analysis.
3. Authors should specify if the examiners were calibrated and in which way.
Our response:
Before evaluation examiners discussed in detail the rules of radiographic healing based on criteria by Rud, Molven and modified PENN 3D for their calibration. The assessment was performed independently, and the inter-examiner agreement of quantitive variables was assessed by calculating the interclass correlation coefficient (ICC2) for absolute agreement with a 2-way random effects, and - in the case of qualitative variables - with the Cohen kappa statistic.
Revised text:
Following the discussion of the criteria for calibration of assessment, radiographic healing was evaluated independently by 2 examiners (D.B and J.W) and classified as: complete, incomplete, uncertain or unsatisfactory healing [33, 35, 36].
- Table 2: Explain the meaning and the abbreviation in table capture
Our response:
It now has been corrected and specified.
Revised text:
M, mean, SD, standard deviation
- Authors should better describe Table 3
Our response:
The table has now been better described.
Revised text:
Table 3. Distribution of cases per preoperative demographic and clinical data as well as radiographic factors with the success rate; univariate logistic regression model.
- Table 5: No Asterisk was used. Please Correct
Our response:
It has now been corrected.
- Explain better this concept “prevent from apical migration of the junctional epithelium”
GTR prevents unwanted cell types such as epithelial cells from the area of regeneration, as the membrane acts as a „barrier” for junctional epithelium formation migrating into the periodontal pocket.
Our response:
It now has been corrected and specified.
Revised text:
Even so, we believe that it is worth trying to treat such lesions, especially with the use of GTR, and although we failed to demonstrate its significant impact on the outcome, the current evidence suggest that GTR substantially improves the results of EMS in the case of apicomarginal lesions [27, 44] as it boosts periapical and periodontal healing, increases both the amount of bone and cementum formation, and prevent from apical migration of the junctional epithelium into the periodontal pocket by acting as a barrier for unwanted cells such as epithelial cells [32].
- “Signifficance”
Our response:
It now has been corrected to “significance”
- Authors might add 4-5 pictures of a clinical case (all of them in a single Figure)
Our response:
Pictures of clinical case have been added to the text as one figure (Fig.1) and the sequence of other figures has been updated.
Revised text:
Figure 1. EMS case presentation (a-e): (a) exposure of the periapical lesion following elevation of the mucoperiosteal flap; (b) effective debridement of the bone defect with the use of Piezosurgeryâ“¡ (Mectron); (c) filling the root-end cavity with the use of MAP systemâ“¡ (Produits Dentaires SA); (d) filling of the defect with bone substitute graft; (e) resorbable membrane covering the graft